# Diagnosis and Antibiotic Treatment of Urinary Tract Infections in Danish General Practice: A Quality Assessment

**DOI:** 10.3390/antibiotics11121759

**Published:** 2022-12-05

**Authors:** Laura Trolle Saust, Volkert Siersma, Jesper Lykkegaard, Lars Bjerrum, Malene Plejdrup Hansen

**Affiliations:** 1Center for General Practice, Department of Clinical Medicine, Aalborg University, 9220 Aalborg, Denmark; 2Section of General Practice and Research Unit for General Practice, Department of Public Health, University of Copenhagen, 1353 Copenhagen, Denmark; 3Audit Project Odense, Research Unit for General Practice, Department of Public Health, University of Southern Denmark, 5000 Odense, Denmark

**Keywords:** antibiotics, urinary tract infections, general practice, quality indicators, quality assessment

## Abstract

Rational antibiotic treatment of urinary tract infections (UTI) is important. To improve the quality of antibiotic treatment of UTI, it is essential to obtain insight into diagnostic approaches and prescribing patterns in general practice. The aim of this study was to investigate the quality of diagnostics and treatment of UTI in general practice by means of quality indicators (QIs). QIs provide a quantitative measure of quality and are defined by a numerator (the number of patients receiving a specific investigation or treatment) and a denominator (the number of patients included in the quality assessment). For adult patients with suspected UTI, practices registered the following: age, sex, risk factors, symptoms and signs, examinations, diagnosis and treatment. The levels of the QIs were compared with their corresponding standards. Half of the patients diagnosed with lower UTI or pyelonephritis fulfilled the diagnostic criteria for UTI: characteristic UTI symptoms and clear signs of bacteriuria, respectively. Urinalysis was performed for nearly all patients, including patients without characteristic symptoms of UTI. One-fourth of the patients with suspected lower UTI were treated with antibiotics despite no urinalysis and nearly half received antibiotics despite an inconclusive dipstick test. Pivmecillam was the preferred antibiotic. The findings of this study indicate that there is room for improvement in the management of UTI in Danish general practice.

## 1. Introduction

Antibiotic resistance has a major impact on human health worldwide and is mainly driven by the use of antibiotics [1]. About 80% of the total human consumption of antibiotics is prescribed in general practice [2]. In Denmark, general practice is organised into small units. A general practice is most often owned by one or several general practitioners (GPs), and staff members—such as secretaries and nurses—are employed. The GPs act as gatekeepers and first-line providers, meaning that a referral from a GP is required for most specialists and always for hospital treatment (except acute admissions) [3].

The prevalence of urinary tract infections (UTI) varies depending on different risk factors such as age and gender. In Denmark, between 5% and 8% of women aged 40–45 years have UTI on a given day [4]. One of the main indications for antibiotic prescriptions in general practice is UTI, particularly for women and elderly patients [5]. In Denmark, total antibiotic consumption has decreased over the past decade [5]. Still, a high number of antibiotic prescriptions are issued in the elderly population, including a relatively high prescription rate of quinolones [6,7,8]. These findings urge interventions to improve the quality of antibiotic use for UTI. Most research focuses on the amount and type of antibiotics prescribed, with very little on the diagnostic process and what lies behind the decision to prescribe antibiotics [9,10,11]. In Danish general practice, not only GPs but also practice staff attend to patients with suspected UTI, although any antibiotics prescribed have to be approved by a GP. Urinalyses, such as microscopy, dipstick, culture and susceptibility testing, are widely performed in general practice. Results are available while patients are still in the consultation (microscopy and dipstick) or within 24 h (culture and susceptibility testing) [12]. Importantly, it may take up to three days to receive the results when urine samples are sent for analysis at the hospital’s microbiological laboratory. Both patients with lower and upper UTI are treated in general practice. In Danish guidelines, uncomplicated lower UTI are defined as UTI in an adult, in non-pregnant woman, and are regardless of age [13,14,15]. Antibiotic treatments for UTI are mostly prescribed at the index consultation, where the patient presents with the UTI for the first time [16]. To improve the quality of care for patients suspected of having UTI it is important to obtain insight into the quality of current diagnostic approaches and prescribing patterns. For this purpose, relevant quality indicators (QIs) can be applied. QIs are measurable parameters that survey potential quality problems [17]. QIs are an important stimulus in the improvement of antibiotic prescribing [18,19]. Recently, a set of indicators comprising the diagnostic process and antibiotic treatment of patients with suspected UTI was developed for use in general practice [20]. These indicators are based on national guidelines and are validated by a panel of experts. By means of the newly developed QIs, the aim of this study was to investigate the present quality of the diagnostic process and antibiotic treatment of UTI in Danish general practice.

## 2. Results

A total of 101 out of the 970 invited practices participated in the study (10.4% response rate). A total of 3076 consultations with patients suspected of having UTI were registered.

At the index consultation, 1514 patients were labelled as “unresolved” (continued suspicion of UTI, but, for example, awaiting a urine culture result). A total of 1263 patients were diagnosed with lower UTI and 16 patients with upper UTI (pyelonephritis). The remaining 219 patients were labelled as “not UTI”, and 64 had a missing diagnosis.

Table 1 shows levels and standards for QIs focusing on the diagnostic process.

Two out of the eleven indicators were within the standard. Nearly all the patients suspected of having UTI who did not have any UTI symptoms (dysuria, frequency, urgency, new-onset incontinence, suprapubic pain or flank pain) nor any systemic signs of infection (fever, shivering or systemically unwell), nevertheless, had a urinalysis performed (urinary dipstick, microscopy and/or urine culture) (QI1). Almost 90% of patients diagnosed with lower UTI had at least one lower urinary tract symptom (QI3). Nonetheless, only about half of the patients diagnosed with either lower UTI or pyelonephritis had UTI symptoms and clear signs of bacteriuria (dipstick positive nitrite and leucocytes and/or positive microscopy) (QI2 and QI5). Most patients diagnosed with complicated lower UTI or pyelonephritis had a urine culture and susceptibility test performed (QI4 and QI6).

Table 2 shows the levels and standards for QIs focusing on the treatment decision.

One out of the six QIs was within the standard. A total of 26% of patients with suspected lower UTI were treated with antibiotics despite the lack of a urinalysis (QI12). Among patients with UTI symptom(s), 11% of those with a negative urinary dipstick (negative leucocytes, negative nitrite) and nearly half of those with an inconclusive urinary dipstick (positive leucocytes, negative nitrite) were treated with antibiotics (QI13 and QI14).

Table 3 shows the levels and standards for QIs focusing on the choice of antibiotics.

Four out of six QIs were within the standard. A total of 83% of patients without a penicillin allergy treated with antibiotics were prescribed pivmecillinam (QI19). Pivmecillinam was also the most frequently prescribed antibiotic for pregnant women and patients diagnosed with pyelonephritis (QI21 and QI24). Very few patients were treated with ciprofloxacin (QI20 and QI23).

## 3. Discussion

### 3.1. Main Findings

According to the diagnostic process, nearly all patients had a urinalysis performed at the index consultation. However, inappropriately, this also included most patients presenting to general practice with no UTI symptoms (QI1). Only about half of the patients who were diagnosed with lower UTI or pyelonephritis fulfilled the two diagnostic criteria: having the characteristic UTI symptoms and a urinalysis indicating bacteriuria, respectively (QI2 and QI5). Regarding the treatment decision, one-fourth of patients with suspected lower UTI were treated with antibiotics despite the lack of a urinalysis (QI12). Nearly half of the patients with lower urinary tract symptom(s) were treated with antibiotics even though they had an inconclusive urinary dipstick (positive leucocytes, negative nitrite) (QI14). The majority of patients treated with antibiotics were prescribed pivmecillinam, and very few were prescribed ciprofloxacin (QI19 and QI20).

### 3.2. Strengths and Limitations

To our knowledge, this is the first study to investigate the quality of the diagnostic process and antibiotic treatment of UTI in general practice by means of validated Qis. Another important strength is the large number of patients and practices included in the study. The data collection process was simple and fast, thus securing the continuous inclusion of patients.

The healthcare professionals who volunteered to register their consultations may have been particularly interested in the management of patients with UTI and may have been more likely to follow guidelines compared to non-participants. Hence, quality may look better in the present data than in Danish general practice in general, which merely underlines the deviations from quality standards found in the present study.

A total of 23 out of the 24 validated QIs were applicable to the data collected using the registration chart. As information about changing the catheter before the start of antibiotic treatment was not available in the APO registration chart, QI17 could not be applied.

The construction of two of the indicators might not sufficiently reflect the widespread use of microscopy in Danish general practice (QI13 and QI14). Microscopic verification of uropathogenic bacteria in a patient with urinary tract symptoms may have led to antibiotic prescribing in spite of a negative dipstick. However, when removing patients with a positive microscopy from the two indicators above (QI13 and QI14), we still found a substantial number of patients treated with antibiotics in spite of a negative or inconclusive dipstick.

Only data from the index consultation were collected. Consequently, we were not able to investigate the quality of any antibiotic treatment prescribed after receiving the results of urine culture and/or susceptibility testing. However, most antibiotics are prescribed on day one [16]. Thus, insight restricted to the care at day one provides an adequate basis for assessing the most central elements in the quality of care for patients with suspected UTI.

### 3.3. Comparisons with Other Studies

#### 3.3.1. QIs Focusing on the Diagnostic Process

A total of 476 patients (15.5%) included in the study had none of the genitourinary symptoms attributable to UTI, nor any systemic signs of infection. Nevertheless, nearly all of these patients had a urinalysis performed with either a dipstick, microscopy and/or urine culture (QI1). Similar results were obtained in another recent Danish study, which found that the majority of patients without UTI symptoms had a urine culture performed [21]. According to the Danish guidelines it is not recommended to perform a urinalysis in patients with the absence of signs and symptoms of UTI [13,14,15,22,23]. Bacteriuria might be misinterpreted as UTI, leading to overprescribing of antibiotics in asymptomatic patients. Despite clear evidence against the treatment of asymptomatic bacteriuria [24], this condition continues to be a common reason for antibiotic treatment [25]. Unnecessary urinalyses promote inappropriate antibiotic prescribing.

UTI are defined by the presence of UTI symptoms in combination with a significant amount of uropathogenic bacteria in the urine [13].

We found that only half of the patients fulfilled the two diagnostic criteria for having UTI: the presence of UTI symptoms and clear signs of bacteriuria (dipstick showing positive nitrite and positive leucocytes and/or positive microscopy) when diagnosed with lower UTI (QI2). Similar results for patients diagnosed with pyelonephritis were found (QI5), indicating a certain degree of mislabelling of both upper and lower UTI.

#### 3.3.2. QIs Focusing on the Treatment Decision

Time pressure, consultation over the phone, and patient treatment expectations are possible explanations for antibiotic prescribing without the result of a preceding urinalysis. These factors are identified as some of the barriers to appropriate antibiotic prescribing [26]. Only about 50% of patients presenting to general practice with urinary tract symptoms actually have UTI [14]. Consequently, treatment based on symptoms alone might lead to the overuse of antibiotics. One of the three main recommendations in the Danish guideline is “no antibiotics without urinalysis” [14]. In this present study, we found that 26% of patients with suspected lower UTI were treated with antibiotics without having a urinalysis performed (QI12).

We found that 11% of patients with lower urinary tract symptoms were treated with antibiotics despite having a negative dipstick (leucocyte and nitrite) (QI13). The probability of having UTI for these patients is low [14,27].

A total of 42% of the patients with the symptom(s) of lower UTI who had an inconclusive urinary dipstick (positive leucocytes and negative nitrite) were treated with antibiotics (QI14). Little et al. found a 75% probability of having UTI when only leucocytes are positive on the dipstick [27]. Holm suggests that the probability is even lower in Denmark [28]. Danish guidelines recommend not initiating antibiotic treatment empirically for patients with an inconclusive urinary dipstick, but waiting for the result of a urine culture [13,14,23].

Nearly all patients diagnosed with pyelonephritis were treated with antibiotics (patients referred to hospitals were excluded) (QI15). This indicates that when GPs suspect “pyelonephritis” they prescribe antibiotics. However, only half of the patients diagnosed with pyelonephritis presented typical symptoms of pyelonephritis and bacteriuria (QI5). This indicates mislabelling leading to the overprescribing of antibiotics. Nevertheless, the underprescribing of antibiotics for pyelonephritis may also take place. Less than half of catheter users with the symptom(s) of pyelonephritis were treated with antibiotics (QI16). In line with our findings, Sommer-Larsen et al. showed that in a nursing home, only about one-third of patients with symptoms indicative of pyelonephritis received antibiotics [21].

#### 3.3.3. QIs Focusing on the Choice of Antibiotics

A total of 83% of patients with suspected lower UTI and no penicillin allergy treated with antibiotics were treated with a first-choice antibiotic (pivmecillinam) (QI19). In line with our findings, a study from 2019 describing the prescriptions for UTI in general practice in Denmark found that pivmecillinam was the most common antibiotic prescribed for lower UTI (45.8% of the prescriptions) [6]. Interestingly, the percentage of patients treated with a first-choice antibiotic for UTI according to national guidelines varies quite a lot in other countries: Belgium (22%), the Netherlands (73%) and Sweden (87%) [29].

We found a very small proportion (1%) of quinolones prescribed for lower UTI (QI20). In comparison with other countries, higher quinolone prescription rates are seen: 22% in Belgium, 7% in the Netherlands, 3% in Sweden and 37% in Switzerland [29,30].

## 4. Materials and Methods

This study is part of a larger quality improvement project originating from Audit Project Odense (APO) with the overall aim of improving the quality of the diagnosis and treatment of UTI in general practice [31,32]. All general practices in the Capital and Southern Regions of Denmark were invited to participate in the project. Throughout a 30-day registration period in January to March 2020 (Capital Region) and October to November 2020 (Region of Southern Denmark), healthcare professionals registered all patients aged ≥18 years with suspected UTI consulting general practice. The suspicion of UTI could have different possible origins, either from the patient, the GP, the general practice staff, a home care nurse and/or a relative. Only information concerning the first day of contact with the general practice (index consultation) was included.

The registrations were performed on A4-size paper registration charts. For each patient, information about the patient’s age, sex, risk factors, symptoms and signs, examinations carried out, diagnosis given and actions made (e.g., hospitalisation, antibiotics prescribed) were recorded in the registration chart (Appendix A).

### 4.1. Quality Indicators

A set of 23 out of 24 validated QIs for the management of patients aged ≥18 years with suspected UTI was applied to the APO dataset [20]. An indicator provides a quantitative measure of quality and is defined by a numerator (the number of patients receiving a specific investigation or treatment) and a denominator (the number of patients included in the quality assessment). Each QI was developed with an accompanying standard to encompass the optimal performance addressed by that indicator. The indicators were divided into three quality domains, focusing on either the diagnostic process, the treatment decision or the choice of antibiotics prescribed. The indicators comprise lower UTI (complicated and uncomplicated), pyelonephritis and some concern only specific patient groups, such as catheter users and pregnant women.

### 4.2. Statistical Analysis

Data from the registration chart were used to calculate the levels of the QIs. The proportion of relevant observations fulfilling each of the 23 QIs was calculated as a percentage and compared with its corresponding standard. A 95% confidence interval (95%CI) for the corresponding odds was estimated using the method of generalised estimating equations (GEE) to adjust for correlation within practices; these were transformed back with the logit formula to obtain the 95%CI for the percentages. The data were analysed using SAS 9.4.

## 5. Conclusions

This study presents an analysis of the quality of care for patients suspected of having UTI in Danish general practice. Urinalyses, such as dipstick, microscopy, culture and susceptibility testing, were found to be used excessively. The findings also indicate antibiotic overuse. However, high adherence to guidelines was found regarding the choice of antibiotic prescribed. Importantly, antibiotic underprescribing was also implied in some cases.

The results of the study are based on the application of a set of QIs. Though it is important to keep in mind that QIs are used to generate reflection and debate about the quality of care rather than a definitive judgment of quality.

The QIs applied in this study represent a useful tool for evaluating and comparing the quality of the diagnosis and antibiotic treatment of patients presenting with suspected UTI.

The findings of the study can be used as a basis for future interventions to improve the quality of management of patients with suspected UTI. Further research should focus on the application of the indicators on a larger scale as part of a system-wide quality investigation program accessible to all general practices in Denmark. The application of the indicators to general practices in, for example, the Nordic countries may also offer an important opportunity to improve care.

## Figures and Tables

**Table 1 antibiotics-11-01759-t001:** Levels and standards for eleven quality indicators focusing on the diagnostic process of urinary tract infections in Danish general practice (2020).

Number of and Rationale behind Quality Indicators	Definition of Quality Indicators ^1^	Patients ^1^ (n:n)	Levels (95% CI) (%)	Standard (%)
Patients with suspected UTI				
QI1: No urinalysis when lack of symptoms	Patients ^a^ without symptoms ^b,e^ who had urinary dipstick and/or microscopy and/or urine culture:Patients ^a^ without symptoms ^b,e^	475:476	99.8 (98.5–100.0)	0–10
Patients with lower UTI				
QI2: UTI is defined by symptoms and bacteriuria	Patients ^g^ with ≥one symptom ^b^ and a positive urinary dipstick (nitrite and leukocytes) and/or a positive microscopy:Patients ^g^	600:1211	49.5 (42.8–56.3)	80–100
QI3: The diagnosis of UTI presupposes symptoms	Patients ^g^ with ≥one symptom ^b^:Patients ^g^	1041:1211	86.0 (81.4–89.6)	90–100
QI4: Urine culture and susceptibility testing when complicated UTI	Patients ^c^ who had urine culture and susceptibility testing:Patients ^c^	301:362	83.1 (75.6–88.7)	90–100
Patients with pyelonephritis				
QI5: Pyelonephritis is defined by symptoms and bacteriuria	Patients ^f^ with ≥one symptom ^e^ and a positive urinary dipstick (nitrite and leukocytes) and/or a positive microscopy:Patients ^f^	8:16	50.0 (27.2–72.7)	90–100
QI6: Urine culture and susceptibility testing when pyelonephritis	Patients ^d^ with ≥one symptom ^e^ who had urine culture and susceptibility testing:Patients ^d^	175:242	72.3 (63.9–79.4)	80–100
QI7: CRP testing can support the diagnosis of pyelonephritis	Patients ^f^ examined with a CRP test:Patients ^f^	15:16	93.8 * (66.1–99.1)	80–100
QI8: Often markedly elevated CRP when pyelonephritis	Patients ^f^ with a CRP test < 20 mg/L:Patients ^f^ examined with a CRP test	8:15	53.3 (26.7–78.2)	0–10
Patients with catheter and UTI				
QI9: Other symptoms in catheter users with UTI	Patients ^h^ with ≥one symptom ^e^:Patients ^h^	26:74	35.1 (24.5–47.5)	90–100
QI10: Urine culture and susceptibility testing for catheter users with UTI	Patients ^h^ who had urine culture and susceptibility testing:Patients ^h^	60:74	81.1 (68.8–89.5)	90–100
Patients with pregnancy and UTI				
QI11: Urine culture and susceptibility testing for pregnant women with UTI	Patients ^i^ who had urine culture and susceptibility testing:Patients ^i^	125:137	91.2 * (84.1–95.3)	90–100

95% CI = 95% confidence interval. A 95% confidence interval for the corresponding odds was estimated in a logistic regression model using the method of generalised estimating equations to adjust for correlation within practices. UTI = urinary tract infection. Catheter = chronic indwelling urethral or suprapubic catheter. * Within the standard. ^1^ Presented as numerator:denominator. ^a^ Patients with suspected UTI (the diagnosis: “complicated lower UTI”, “uncomplicated lower UTI”, “pyelonephritis”, “unresolved” and “other, not UTI”). ^c^ Patients diagnosed with complicated lower UTI (the diagnosis: “complicated lower UTI”) including catheter users and pregnant women. ^d^ Patients with suspected pyelonephritis (the diagnosis: “pyelonephritis” and “unresolved”). ^e^ Symptoms of pyelonephritis: fever, shivering, flank pain, systemically unwell. ^f^ Patients diagnosed with pyelonephritis (the diagnosis: “pyelonephritis”). ^g^ Patients diagnosed with lower UTI (the diagnosis: “complicated lower UTI” and “uncomplicated lower UTI”). ^h^ Catheter users with suspected UTI (the diagnosis: “complicated lower UTI”, “uncomplicated lower UTI”, “pyelonephritis” and “unresolved”). ^I^ Pregnant women with suspected UTI (the diagnosis: “complicated lower UTI”, “uncomplicated lower UTI”, “pyelonephritis” and “unresolved”).

**Table 2 antibiotics-11-01759-t002:** Levels and standards for seven quality indicators focusing on the treatment decision for urinary tract infections in Danish general practice (2020).

Number of and Rationale behind Quality Indicators	Definition of Quality Indicators ^1^	Patients ^1^ (n:n)	Levels (95% CI) (%)	Standards (%)
Patients with lower UTI				
QI12: No antibiotics without urinalysis	Patients ^a^ not examined with a urinary dipstick or microscopy treated with antibiotics:Patients ^a^ not examined with a urinary dipstick or microscopy	13:51	25.5 (14.7–40.4)	0–10
QI13: No antibiotics for patients with low probability of bacteriuria	Patients ^a^ with ≥one symptom ^b^ and a negative urinary dipstick (nitrite and leukocytes) treated with antibiotics:Patients ^a^ with ≥one symptom ^b^ and a negative urinary dipstick (nitrite and leukocytes)	46:406	11.3 (8.4–15.1)	0–10
QI14: Wait with antibiotics when inconclusive urinary dipstick	Patients ^a^ with ≥one symptom ^b^ and a urinary dipstick with positive leukocytes but negative nitrite treated with antibiotics:Patients ^a^ with ≥one symptom ^b^ and a urinary dipstick with positive leukocytes but negative nitrite	444:1061	41.8 (36.1–47.8)	0–20
Patients with pyelonephritis				
QI15: Antibiotics for patients with pyelonephritis	Patients ^c^ treated with antibiotics:Patients ^c^	13:14	92.9 * (62.5–99.0)	90–100
Patients with catheter and UTI				
QI16: Antibiotics for catheter users with UTI	Patients ^e^ with ≥one symptom ^d^ treated with antibiotics:Patients ^e^ with ≥one symptom ^d^	11:26	42.3 (28.1–58.0)	90–100
QI17 ^2^: Change of catheter for catheter users with UTI	Patients ^e^ with ≥one symptom ^d^ and change of catheter:Patients ^e^ with ≥one symptom ^d^	-	-	90–100
Patients with pregnancy and UTI				
QI18: No antibiotics for pregnant women without urinalysis	Patients ^f^ not examined with a urinary dipstick or microscopy treated with antibiotics:Patients ^f^ not examined with a urinary dipstick or microscopy	1:4	25.0 (3.4–76.2)	0–10

95% CI = 95% confidence interval. A 95% confidence interval for the corresponding odds was estimated in a logistic regression model using the method of generalised estimating equations to adjust for correlation within practices. UTI = urinary tract infection. Catheter = chronic indwelling urethral or suprapubic catheter. * Within the standard. ^1^ Presented as numerator:denominator. ^2^ Data not available, the QI is excluded. ^a^ Patients with suspected lower UTI (the diagnosis: “complicated lower UTI”, “uncomplicated lower UTI” and “unresolved”). ^b^ Symptoms of lower UTI: dysuria, frequency, urgency, new-onset incontinence, suprapubic pain. ^c^ Patients diagnosed with pyelonephritis (the diagnosis: “pyelonephritis”) excluding patients referred to hospital. ^d^ Symptoms of pyelonephritis: fever, shivering, flank pain, systemically unwell. ^e^ Catheter users with suspected UTI (the diagnosis: “complicated lower UTI”, “uncomplicated lower UTI”, “pyelonephritis” and “unresolved”) excluding patients referred to hospital. ^f^ Pregnant women with suspected UTI (the diagnosis: “complicated lower UTI”, “uncomplicated lower UTI”, “pyelonephritis” and “unresolved”).

**Table 3 antibiotics-11-01759-t003:** Levels and standards for six quality indicators focusing on the choice of antibiotics prescribed for urinary tract infections in Danish general practice (2020).

Number of and Rationale behind Quality Indicators	Definition of Quality Indicators ^1^	Patients ^1^ (n:n)	Levels (95% CI) (%)	Standards (%)
Patients with lower UTI				
QI19: Pivmecillinam is first-choice antibiotic for treatment of UTI	Patients ^a^ with no penicillin allergy treated with pivmecillinam:Patients ^a^ with no penicillin allergy treated with antibiotics	733:886	82.7 (79.2–85.8)	90–100
QI20: Ciprofloxacin is not first-choice antibiotic for treatment of lower UTI	Patients ^a^ treated with ciprofloxacin:Patients ^a^ treated with antibiotics	10:906	1.1 * (0.6–2.0)	0–5
Patients with pyelonephritis				
QI21: Pivmecillinam is first-choice antibiotic for treatment of pyelonephritis	Patients ^b^ with no penicillin allergy treated with pivmecillinam:Patients ^b^ with no penicillin allergy treated with antibiotics	13:14	92.9 * (66.1–98.9)	90–100
QI22: Ciprofloxacin for treatment of pyelonephritis only if penicillin allergy	Patients ^b^ with penicillin allergy treated with ciprofloxacin:Patients ^b^ treated with ciprofloxacin	0:0	-	90–100
Patients with catheter and UTI				
QI23: Ciprofloxacin is not first-choice antibiotic for treatment of catheter users with UTI	Patients ^c^ with no penicillin allergy treated with ciprofloxacin:Patients ^c^ with no penicillin allergy treated with antibiotics	1:23	4.3 * (0.8–20.7)	0–10
Patients with pregnancy and UTI				
QI24: Pivmecillinam is first-choice antibiotic for treatment of pregnant women with UTI	Patients ^d^ with no penicillin allergy treated with pivmecillinam:Patients ^d^ with no penicillin allergy treated with antibiotics	22:24	91.7 * (72.0–97.9)	90–100

95% CI = 95% confidence interval. A 95% confidence interval for the corresponding odds was estimated in a logistic regression model using the method of generalised estimating equations to adjust for correlation within practices. UTI = urinary tract infection. Catheter = chronic indwelling urethral or suprapubic catheter. * With 95% possibility within the standard. ^1^ Presented as numerator:denominator. ^a^ Patients with suspected lower UTI (the diagnosis: “complicated lower UTI”, “uncomplicated lower UTI” and “unresolved”). ^b^ Patients diagnosed with pyelonephritis (the diagnosis: “pyelonephritis”). ^c^ Catheter users with suspected UTI (the diagnosis: “complicated lower UTI”, “uncomplicated lower UTI”, “pyelonephritis” and “unresolved”). ^d^ Pregnant women with suspected UTI (the diagnosis: “complicated lower UTI”, “uncomplicated lower UTI”, “pyelonephritis” and “unresolved”).

## Data Availability

The data presented in this study are not publicly available due to privacy restrictions.

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
