# Peer review of "Diagnosis and Antibiotic Treatment of Urinary Tract Infections in Danish General Practice: A Quality Assessment"

_antibiotics, 2022, doi:10.3390/antibiotics11121759_

Round 1

Reviewer 1 Report

I have read with interest the manuscript by Saust et al, since AMR represents a major concern worldwide.

I want to congratulate the authors for the present work and the quality of the presentation, it represents valuable research for this field.

*row 71 - Some 1263 patients

Author Response

Point-to-point letter

Reviewer 1:

(1) I have read with interest the manuscript by Saust et al, since AMR represents a major concern worldwide.

I want to congratulate the authors for the present work and the quality of the presentation, it represents valuable research for this field.

*row 71 - Some 1263 patients

Answer:

Thank you very much for your positive comments. We have changed the sentence in the revised manuscript to: “A total of 1263 patients...”

Reviewer 2 Report

1) Change the word threat in line number 32.

2) Globally and specifically in Danish society, the prevalence of UTI is missing.

3) Study design. sampling technique and calculation of sample is missing.

4) Consent form is also missing.

5) How to validate the questionnaire QI? 

6) Ethical approval is missing.

7) QI of antibiotics is still missing in paper? rearrange the title or add some more results.

8) Results are insufficient in terms of antibiotics.

9) Methodology should be discussed in detail.

Author Response

Reviewer 2:

(1) Change the word threat in line number 32.

Answer:

Thank you very much. We have changed the sentence in the revised manuscript to: “Antibiotic resistance has a major impact on human health worldwide and is mainly driven by the use of antibiotics.”

(2) Globally and specifically in Danish society, the prevalence of UTI is missing.

Answer:

We agree that this is important information to include. We have added the following sentence in the revised manuscript: “The prevalence of urinary tract infections (UTI) varies - depending on different risk factors such as for example age and gender. In Denmark, between 5% and 8% of women aged 40-45 years has a UTI at a given day”.

(3) Study design. sampling technique and calculation of sample is missing.

Answer:

Thank you very much. The research team agrees that the study design should be elaborated.

Three sections on study design have been added. “The registrations were performed on A4 size paper registration charts”, has been added to the Materials and Methods section. Further: “An indicator provides a quantitative measure of quality and is defined by a numerator (number of patients receiving a specific investigation or treatment) and a denominator (number of patients included in the quality assessment)”, has been added to the Quality indicators section and finally: “Observations in the registration chart were used to calculate the levels of the QIs”, has been added in the Statistical analysis section. Please see revised manuscript.

(4) Consent form is also missing.

Answer:

According to Danish law a patient consent form is not required for this type of study. Data are fully anonymous, as no personal identifiable information such as name or Central Person Registration (CPR) number was obtained. All healthcare participants gave their informed written consent. We have changed the Institutional Review Board Statement section to: “All healthcare participants consented to the study. Only anonymised patient data were obtained. The project is registered at the Research Unit for General Practice in Aalborg (ID:251-3).” Please see revised manuscript.

(5) How to validate the questionnaire QI?

Answer:

We agree that this is important information. The QIs have face and content validity. A section: “These indicators are based on national guidelines and are validated by a panel of experts” is added to the information section. Please see revised manuscript.

(6) Ethical approval is missing.

Answer:

No approval from an ethics committee was obtained since this is not required for Danish register-based studies. Please see the Institutional Review Board Statement section in the revised manuscript.

(7) QI of antibiotics is still missing in paper? rearrange the title or add some more results.

Answer:

This study comprises results of both the diagnostic process and antibiotic treatment of UTIs.
Out of the total 24 QIs some six indicators focused on the decision to treat with antibiotics (Table 2) and another six on the choice of antibiotics (Table 3).
Please see Table 2 and 3 in the revised manuscript.

(8) Results are insufficient in terms of antibiotics.

Answer:
Please see answer above.

The set of quality indicators used for this study is mainly developed for use in a Danish setting. Consequently, the indicators only focus on the type of antibiotics mainly used in Denmark for treatment of UTIs. Importantly, both Table 2 and 3 includes information about use of antibiotics.

(9) Methodology should be discussed in detail.

Answer:

Thank you for your comment. Please see the sections added in the Materials and Methods section in the revised manuscript.

Reviewer 3 Report

In Abstract, the methodology should be described more clearly.

In Introduction you state not only GPs but also often practice staff attend patients with suspected UTI - can you please describe in greater detail who can diagnose and manage UTI in Denmark, i.e. are pharmacists also able to dispense antibiotics?

Is delayed prescribing an option in Denmark?

Provided supplementary and nonpublished files are the same, please check.

Why 23 of 24 QI, provide rationale. Which one was excluded as one of the tables states QI24

Data on statistical analysis should be provided in table footnote

Author Response

Point-to-point letter

Reviewer 3:

(1) In Abstract, the methodology should be described more clearly.

Answer:

Thank you very much for the suggestion. We have added a description of the methodology to the abstract section: “QIs provide a quantitative measure of quality and is defined by a numerator (number of patients receiving a specific investigation or treatment) and a denominator (number of patients included in the quality assessment)”. Please see revised manuscript. 

(2) In Introduction you state not only GPs but also often practice staff attend patients with suspected UTI - can you please describe in greater detail who can diagnose and manage UTI in Denmark, i.e. are pharmacists also able to dispense antibiotics?

Answer:

This is a relevant comment and we understand your confusion. In Denmark only GPs can prescribe antibiotics. Nevertheless, often practice staff attend patients with acute infections in general practice. They diagnose and suggest antibiotic prescriptions, which then must be approved by the GP before it is dispensed. Pharmacists are able to dispense antibiotic when prescribed by the GPs. We changed the sentence in the introduction section: “In Danish general practice, not only GPs but also practice staff attend patients with suspected UTI, although any antibiotics prescribed has to be approved by a GP”, and added: “A general practice is most often owned by one or several general practitioners (GPs) and staff members – such as secretaries and nurses – are employed”. Please see revised manuscript.

(3) Is delayed prescribing an option in Denmark?

Answer:

Yes, delayed antibiotic prescribing is an option in Denmark. Some GPs make use of it while others do not endorse it. This study solely focused on antibiotics prescribed, and does not account for the amount of delayed prescribing.

(4) Provided supplementary and nonpublished files are the same, please check.

Answer:

Thank you for your observation. There should only be one supplementary appendix A and no nonpublished files. It has been corrected.

(5) Why 23 of 24 QI, provide rationale. Which one was excluded as one of the tables states QI24

Answer:

One QI, QI17, could not be calculated because we did not have information about change of catheter. In the Discussion, Strengths and limitations section it is stated: “A total of 23 out of the 24 validated QIs were applicable to the data collected using the registration chart. As information about changing catheter before startup of antibiotic treatment was not available in the APO registration chart, QI17 could not be applied.” For QI17 we added information in the footnote 2: “Data not available, the QI is excluded.” Please see Table 2 in the revised manuscript.

(6) Data on statistical analysis should be provided in table footnote

Answer:

Thank you. The 95% confidence intervals (CI) are provided in brackets after the levels in Table 1, 2 and 3. We have added information on statistical analysis for 95% CI in the footnotes of the three tables: 95% CI = 95% Confidence interval. A 95% confidence intervals for the corresponding odds were estimated in a logistic regression model using the method of generalised estimating equations to adjust for correlation within practices”. Please see revised Table 1, 2 and 3.

Round 2

Reviewer 2 Report

Revised version is ok.